# Optical Coherence Tomography Angiography as a Diagnostic Tool for Diabetic Retinopathy

**DOI:** 10.3390/diagnostics14030326

**Published:** 2024-02-02

**Authors:** Naomi Wijesingha, Wei-Shan Tsai, Ayse Merve Keskin, Christopher Holmes, Dimitrios Kazantzis, Swati Chandak, Heena Kubravi, Sobha Sivaprasad

**Affiliations:** 1UCL Institute of Ophthalmology, London EC1V 9EL, UK; sobha.sivaprasad@nhs.net; 2Moorfields Eye Hospital, London EC1V 2PD, UK; wei-shan.tsai@nhs.net (W.-S.T.); a.keskin@nhs.net (A.M.K.); christopher.holmes11@nhs.net (C.H.); dimitrios.kazantzis@nhs.net (D.K.); swati.chandak@nhs.net (S.C.); syed.kubravi@nhs.net (H.K.)

**Keywords:** artefacts, choriocapillaris, diabetic macular ischaemia, diabetic macular oedema, diabetic retinopathy, fundus fluorescein angiography, non-proliferative diabetic retinopathy, optical coherence tomography angiography, pre-diabetic, proliferative diabetic retinopathy

## Abstract

Diabetic retinopathy (DR) is the most common microvascular complication of diabetes mellitus, leading to visual impairment if left untreated. This review discusses the use of optical coherence tomography angiography (OCTA) as a diagnostic tool for the early detection and management of DR. OCTA is a fast, non-invasive, non-contact test that enables the detailed visualisation of the macular microvasculature in different plexuses. OCTA offers several advantages over fundus fluorescein angiography (FFA), notably offering quantitative data. OCTA is not without limitations, including the requirement for careful interpretation of artefacts and the limited region of interest that can be captured currently. We explore how OCTA has been instrumental in detecting early microvascular changes that precede clinical signs of DR. We also discuss the application of OCTA in the diagnosis and management of various stages of DR, including non-proliferative diabetic retinopathy (NPDR), proliferative diabetic retinopathy (PDR), diabetic macular oedema (DMO), diabetic macular ischaemia (DMI), and pre-diabetes. Finally, we discuss the future role of OCTA and how it may be used to enhance the clinical outcomes of DR.

## 1. Introduction

Diabetes Mellitus (DM) is a chronic, progressive, multiorgan disease, representing one of the fastest-growing global health problems of the 21st century [1]. The International Diabetes Federation reported that, in 2021, DM affected 537 million individuals worldwide, a global prevalence of over 10% of the population, with an estimated increase to 783 million by 2045 [1]. The complications of DM, both macrovascular and microvascular, are not only a public health burden but can cause physical, emotional, and financial burdens at an individual level [2,3].

Diabetic retinopathy (DR) is the most common microvascular consequence of DM, with a recent meta-analysis of prevalence data from population-based studies conducted around the world highlighting that up to a third of people with diabetes have DR [4]. Although preventable, DR remains a prominent cause of acquired visual impairment in working-age adults across the globe, with an estimated 103 million having DR and nearly 30 million suffering from vision-threatening diabetic retinopathy (VTDR) [4]. VTDR comprises proliferative diabetic retinopathy (PDR) and diabetic macular oedema (DMO).

Traditionally, the diagnosis and management of DR relied upon dilated ophthalmoscopy, colour fundus photography (CFP), fundus fluorescein angiography (FFA) and optical coherence tomography (OCT) [5]. More recently, optical coherence tomography angiography (OCTA) has provided high-resolution imagery of the clinically relevant retinal vasculature of DR, further enhancing our understanding of DR [6].

OCTA is a fast, non-invasive, contactless technique that enables the visualisation of the retinal vasculature by tracking the movement of red blood cells over time [7]. Currently, there is conflicting evidence regarding changes in retinal blood flow in early DR, with some studies suggesting increased retinal blood flow [8,9] and others indicating decreased retinal blood flow in early DR [10,11]. Furthermore, considering that neuronal degeneration and vasoregression in the diabetic retina precede observable DR [12,13], OCTA has the potential to detect DR before it can be diagnosed through traditional fundus examinations, emphasising the valuable role OCTA has in advancing our understanding of DR.

This review will focus on the current applications of OCTA as a diagnostic tool for diagnosing and managing DR.

## 2. Methods

The references were searched on two major research platforms, including PubMed and Medline, using the keywords “artefacts”, “choriocapillaris”, “diabetic macular ischaemia”, “diabetic macular oedema”, “diabetic retinopathy”, “fundus fluorescein angiography”, “non-proliferative diabetic retinopathy”, “optical coherence tomography angiography”, “pre-diabetic”, and “proliferative diabetic retinopathy”, with a mixture of explode and MeSH term function. Search strategies, such as truncation and wildcard symbols, were employed alongside the Boolean Operators when applicable. We set the search filter starting from the year 2014 when OCTA was first commercially available. The abstracts were screened first, and only papers written in English were recruited. A total of 159 articles were selected, reviewed, and formed the major component of this manuscript.

## 3. Principles of Optical Coherence Tomography Angiography

It is essential to understand the basic principles of OCTA and how it enables non-invasive, depth-resolved images of the retinal vasculature [5,6] without the need for pupil dilation [14]. The principle involves capturing consecutive OCT B-scans in the same retinal location whilst monitoring the motion of erythrocytes within the retinal vasculature. Changes in contrast over time indicate the location of the blood vessels. Subsequently, the OCTA device processes these images to create a comprehensive angiogram of the retinal vascular supply [5].

Since its development in 2015, OCTA has undergone several revolutionised improvements, including an increase in scanning speed to optimise the image resolution. Currently, the commercially available OCTA systems mainly adopt two algorithms to form their images: split-spectrum amplitude-decorrelation angiography (SSADA) [15] and optical micro-angiography (OMAG) [16]. There are other algorithms being investigated; however, SSADA is the most well-known one. The SSADA algorithm gains its reputation by decreasing the sensitivity to axial motion noise, thereby improving the signal-to-noise (SNR) ratio. Currently, AngioVue (Optovue) OCTA has an inbuilt SSADA technique, whilst AngioPlex and PLEX Elite 9000 utilise OMAG algorithm. These algorithms all aim at helping clinicians to visualise the retinal vasculature clearly.

Understanding the retinal vasculature is essential to appreciate how OCTA aids in the clinical assessment of DR. The retina comprises four distinct retinal plexuses: (1) the radial peripapillary capillary plexus (RPCP) is within the retinal nerve fibre layers and emanates from the optic nerve; (2) the superficial capillary plexus (SCP) which supplies the retinal ganglion cell layer and inner plexiform layer; (3) intermediate capillary plexus (ICP) which supplies the inner plexiform layer and inner nuclear layer; and (4) deep capillary plexus (DCP) which predominantly supplies the inner nuclear layer [17]. The capillaries of the SCP terminate at the foveal avascular zone (FAZ), a relatively circular, capillary-free zone at the fovea [18]. The DCP follows similar anatomy around the FAZ; however, the dimensions of the FAZ in the DCP are not as well visualised on OCTA. The choriocapillaris (CC) predominantly supplies the outer retina, yet about 15% of the oxygen supply to the photoreceptors is from the DCP [19]. A significant advantage of OCTA in the study of DR is the capability to dissect the relative vascular changes in each plexus and offer objective markers for quantifying pathological changes [18].

### 3.1. Optical Coherence Tomography Angiography vs. Fundus Fluorescein Angiography

Although FFA remains an important test for staging and monitoring DR in clinical practice, it has its limitations. The invasiveness and time consumption of the test have restricted its clinical usage [20]. The dye injected for FFA can cause nausea and vomiting and, in rare cases, lead to life-threatening anaphylaxis [21]. With its rapid advancements and non-invasive nature, OCTA has several advantages summarised in Table 1, which has prompted numerous studies exploring its role as a diagnostic tool for DR [20,22].

While FFA assesses the retinal superficial vascular changes in DR, the deeper capillary plexuses cannot be appreciated. In contrast, OCTA distinguishes changes to defined retinal layers at the SCP, ICP, DCP, and CC [24,25,26]. Additionally, OCTA offers a superior image resolution compared to FFA, especially in areas of capillary non-perfusion, and has quantitative data processing capabilities to measure various critical parameters, such as vessel density (VD), vessel length density (VLD), perfusion density (PD), and detailed characteristics of the FAZ [27]. Some studies demonstrated significant associations between these metrics, nonperfusion areas (NPA), and the diagnosis and analysis of the severity of DR [28,29]. OCTA also enables a comprehensive and phenotypic assessment of neovascularisation (NV), potentially providing valuable prognostic insights with direct clinical implications [30].

However, it is crucial to recognise that OCTA has some limitations compared to FFA. It provides a limited view of the peripheral retina compared to FFA, although wider field of view techniques are being developed with swept source-OCTA (SS-OCTA) [22,31]. The primary factor contributing to these challenges is the presence of artefacts, which can affect the overall quality of the images [26]. These artefacts, in combination with the time required for acquisition and the automated fusion process of the OCTA device, can affect the extensive scan dimension necessary for creating the OCTA montage of images [22,32]. Consequently, maintaining stable fixation is essential for obtaining high-resolution images with OCTA, which can be particularly demanding for patients with severe macular diseases, including DMO and diabetic macular ischaemia (DMI).

While OCTA has not yet replaced FFA, one significant factor contributing to this is that OCTA cannot directly visualise leakage from blood vessels, limiting its effectiveness in differentiating vascular leakage, staining, or pooling [26,33]. A recent advancement involved the use of artificial intelligence-inferred fluorescein angiography (AI-FA) based on OCTA images [34]. AI-FA demonstrated sufficient quality in delineating leakage in eyes with DR, enabling precise DR diagnosis and treatment planning. At present, FFA and OCTA are complementary tests; however, the ongoing research trajectory aims at replacing FFA with OCTA for the detection and management of DR.

### 3.2. Optical Coherence Tomography Angiography Artefacts

Artefacts are a common feature of OCTA, which may be perceived as a technique limitation. They can be attributed to various causes and are critical to consider when interpreting OCTA images. Artefacts can be broadly categorised into signal artefacts and processing artefacts. The most common artefacts seen are related to motion, shadow, and defocus, but if recognised, they can be improved with operator training [35].

#### 3.2.1. Signal Artefacts

Signal artefacts can be divided into false positive and false negative flow artefacts. As the names imply, false positive artefacts indicate flow in its absence, and false negative artefacts occur when the true flow is undetected. False positive flow can result from image noise, projection, motion artefacts, or suspended scattering particles in motion (SSPiM). False negative flow can be related to low signal, shadow artefacts [36], and low flow motion below a detectable level [37].

Noise in OCTA is non-signal information originating from various sources. Its impact increases as signal strength and signal-to-noise ratio (SNR) diminish. Low signal can be related to numerous factors, including defocus or media opacities [36]. A low SNR is particularly problematic when assessing small capillary networks near the FAZ [38] or deeper retinal tissues like the choroid [39] as the signal deteriorates on passage through normal retinal tissues, making them harder to distinguish from noise. Most systems give signal strength indicators suggesting the likelihood of noise-related artefacts. In addition, SS-OCTA can help image deeper tissues in the choroid [39].

Projection artefacts are common and important to distinguish from true signals. An example of a projection artefact is seen in Figure 1. They occur when light passes through the superficial plexus, and the signals might be absorbed, reflected, or scattered as the blood flows. The resultant fluctuating signals subsequentially change the illumination of the deeper layer, creating a false positive flow artefact [36]. The artefact tends to be more prominent in hyper-reflective structures such as the retinal pigment epithelium (RPE) and plexiform layers. Consequently, projection artefacts can cause deviations in reading results, such as falsely increased VD in the DCP [18]. It can be detected by reviewing the B scan images, where the vertical columns of flow or the reflections of flow at the RPE directly below retinal vessels can be seen.

Motion artefacts are another source of false positive flow due to voluntary or involuntary fixation eye movement that is not flow-related (Figure 2). OCTA detects flow by measuring individual pixel decorrelation or fluctuation on repeated imaging [36], assuming these changes represent flow. *En-face* images show eye movement as easily detectable gaps, shearing, or distortion [36]. Pupil tracking can allow rescanning of the affected areas [40,41,42], and training technicians can help reduce their incidence [35]. Movement within the eye can also create artefacts. The pulsatile choroidal expansion causes *z*-axis movement that is usually corrected, but non-uniform expansion could cause regions of false positive flow. Variable reflection from pathology such as cystoid macular oedema (CMO) or lipid may also create decorrelation that gives a false positive flow signal [36].

SSPiM was first described by Kashani et al., who discovered these extravascular motion signals on OCTA [43]. As its name suggests, SSPiM is associated with motion; however, this imaging feature can be easily distinguished from motion artefacts by its location and morphology (see Figure 2 and Figure 3). Specifically, SSPiM is usually located close to the end of capillaries with an avoid shape. In the same article, the authors designed an experiment where 1% intralipid particles and 1% gelatin mixed solution were imaged by OCTA. The mini flow of those particles in the solution was captured and presented as signals similar to SSPiM. Moreover, the team reported the correlation between SSPiM and hyper-reflective materials in the outer plexiform layer on OCT. Together, Kashani et al. proposed this novel OCTA feature as a biomarker of exudative maculopathy, especially the presence of hard exudate. In our preliminary study, SSPiM falsely increases the vessel density signals and hinders the software from delineating the FAZ correctly (Figure 3).

Decentration artefact is another problematic issue associated with motion artefacts. Although decentration does not generate new false positive signals, it leads the computer software to misinterpret the VD and sometimes the FAZ (Figure 4). Moreover, the smaller field 3 × 3 mm macular scans are more prone to severe decentration artefact than the larger field 6 × 6 mm scans [35]. In our experience, once the computer is fixed at a decentred point, it will continuously use the wrong fixation centre in the following visits. These inaccurate readings often have to be excluded from analysis, which should be prevented at the first visit by setting the fixation centre correctly.

Shadow artefacts (Figure 5) occur where the signal is reduced by overlying material, causing false negative flow. This artefact can also be caused by a relatively transparent material, such as vitreous opacities, vitreous haemorrhage, and inflammatory cell aggregates, preventing signals from deeper tissues [36], and thereby affecting the readings of FAZ and VD. Consequently, researchers cannot evaluate whether there is an actual loss of capillaries. Moreover, scientists have yet to develop a correct algorithm for the false negative flow caused by medial opacities, adding difficulties in making accurate clinical interpretations.

Thresholding in OCTA also contributes to the generation of false negative signals [38]. The purpose of thresholding is to remove signals above or below a certain level, thereby enhancing image visualisation. Consequently, when erythrocytes (the cells responsible for reflecting light emitted from OCTA) move slowly, or there is a decreased number in the smaller capillaries after bifurcation (also known as plasma skimming vessels), the reflected light would be too weak to generate a detectable signal above the threshold, hence the false negative flow [44]. Similarly, one can deduce that the peripheral cell-free layer in larger vessels, where no erythrocytes are present, could result in a smaller calibre vessel on the image display because of the false negative effect [45]. All these signal artefacts have to be considered when interpreting OCTA results.

#### 3.2.2. Processing Artefacts

Processing artefacts are generated by post hoc analysis or the manipulation of the image signal in a way that affects interpretation. These artefacts are most commonly segmentation errors of the retinal architecture but can also be related to artefact compensation methods.

In normal anatomy, automatic segmentation of the retinal layers accurately informs the *en-face* OCTA images. In pathologies such as DMI, DMO, and the disorganisation of retinal layers (DRIL), segmentation is often inaccurate due to the disruption or loss of the retinal layers [7]. This artefact creates errors in the *en-face* images and prevents accurate assessment of retinal layers (Figure 6). Segmentation artefacts can be identified using cross-sectional B scans, and manual correction can be performed, but it is time-consuming [36].

Software-based correction methods exist to compensate for noise, projection, and motion artefacts but are liable to create other artefacts. De-noising can cause false negative flow by removing low signals from small vessels or adding information [46,47], projection correction can cause “dark vessels” of negative flow [48], and motion correction can create quilting and vascular doubling [38].

### 3.3. Spectrum-Domain versus Swept-Source Optical Coherence Tomography Angiography

There are two types of OCTA currently available in the market. One is the spectrum-domain OCTA (SD-OCTA) provided by AngioVue XR Avanti (Optovue, Fremont, CA, USA), Spectralis (Heidelberg Engineering, GmbH, Heidelberg, Germany), Carl Zeiss Cirrus 5000 (Zeiss, Dublin, CA, USA), or Zeiss Cirrus Angioplex (Zeiss, Dublin, CA, USA), and the other one is the swept-source OCTA (SS-OCTA) offered by Zeiss PLEX Elite 9000 (Zeiss, Dublin, CA, USA), or Topcon Triton (Topcon, Tokyo, Japan). SD-OCTA is the first commercialised OCTA device utilising the light of 840 nm with a scanning speed above 68,000 A-scans/s, whilst SS-OCTA comes into the market later and is equipped with a light source of 1050 nm and a scanning speed of at least 100,000 A-scans/s.

Due to the difference in light source and scanning speed, SS-OCTA can penetrate the CC tissue deeper, providing more accurate details underneath the RPE. We know that CC is responsible for nourishing the photoreceptors, and CC flow void might lead to irreversible cell death seen as EZ loss on OCT, resulting in visual compromise in DR.

Outside of DR, several studies have compared the performance of the two devices in detecting various ocular conditions associated with choroidopathy. For example, SS-OCTA could alleviate the shadow artefact caused by subretinal fluid in eyes with acute central serous retinopathy, thereby showing more homogenous decreased CC flow than SD-OCTA [49]. Another study compared SD-OCTA and SS-OCTA’s ability to detect macular neovascularisation (MNV) in cases with subretinal hyper-reflective materials (SHRM) and/or pigment epithelial detachment (PED) on OCT, which showed more sensitivity in SS-OCTA after manual correction of segmentation error [50]. Finally, in the eyes with confirmed choroidal neovascularisation (CNV), SS-OCTA tends to display the lesion with a larger size than SD-OCTA.

A dedicated study comparing SD-OCTA versus SS-OCTA in detecting CC deficit in DR is still lacking, which is worth investing in as a future research direction. In one study on DMO cases, cysts can be hypo- or hyper-reflective in both devices [51]; however, SS-OCTA is more likely to detect hyper-reflective SSPiM, which corresponds to cysts on OCT B-scan. Nevertheless, studies examining CC changes in diabetes using SS-OCTA alone have grown exponentially. Of note, pre-diabetic and early-stage DR patients exhibited a significantly decreased choroidal vascularity index compared to the normal controls, which might serve as a biomarker to detect DR early [52]. Moreover, a one-year study observing 1222 diabetic patients disclosed that a higher CC flow deficit percentage (CC FD%) at baseline predicted the development of referrable DR (i.e., moderate NPDR above or DMO) at one year reliably [53]. The application of OCTA in the early detection of diabetic choroidopathy is promising; however, one should remember that the results in SD-OCTA and SS-OCTA are not always interchangeable [54].

### 3.4. Different Metrics on Optical Coherence Tomography Angiography

OCTA provides a range of quantitative metrics used to assess the vascular structure of the retina and choroid. These have been summarised in Table 2.

Recent studies have explored incorporating these metrics to enhance the existing diagnostic prediction model for DR [26,55,56,57,58]. These metrics provide quantifiable measurements used to assess the microvascular alterations associated with DR. Amongst the commonly reported indices are the expansion of FAZ area, lengthened perimeter, decreased circularity, and reduced VD in SCP and SCP as an indicator of disease progression [59,60,61]. An example of an enlarged FAZ from DM seen on OCTA is shown in Figure 7.

Throughout this review, we will discuss each OCTA metric in relation to the stage of DR analysed and how these might be valuable tools to refine DR diagnosis.

## 4. Optical Coherence Tomography Angiography and Diabetic Retinopathy

The pathophysiological processes underlying DR can be broadly summarised as the loss of the retinal capillary network in both the peripheral and central retina, increased vascular permeability, the onset of inflammation, and neurodegeneration from chronic hyperglycaemia [7]. These result in observable clinical manifestations within the retina, such as the development of microaneurysm, dot and blot haemorrhages, venous beading, and ultimately, the formation of new blood vessels within the retina and optic disc [62].

The gold standard in the classification of DR has historically been the Early Treatment of Diabetic Retinopathy Study (ETDRS) [63]. A more practical approach is the International Clinical Diabetic Retinopathy and Diabetic Macula Edema Severity Scale [64]. This scale categorises retinopathy into five disease categories and differentiates oedema into two categories, alleviating the need for detailed grading and improving diagnostic efficacy in practice.

### 4.1. Clinical Features of Diabetic Retinopathy

For OCTA studies, patients with DM are categorised into three main groups [64]: DM without clinical signs of DR, non-proliferative diabetic retinopathy (NPDR), and proliferative diabetic retinopathy (PDR). On colour fundus photography (CFP), NPDR is further divided into mild, moderate, and severe stages based on specific vascular changes, such as microaneurysms, dot and blot haemorrhages, intraretinal microvascular abnormalities (IRMA), and venous beading. Similarly, PDR is defined by the presence of vitreous haemorrhage and neovascularisation at the disc (NVD) and/or elsewhere (NVE), while DMO is graded absent or present [64].

Although most of these signs can be seen on OCTA, the restricted field of view is a disadvantage for use as a screening tool.

However, the depth-resolved images of OCTA have emerged as a promising tool in providing quantifiable status of retinal capillary microvasculature. As a result, OCTA has greatly advanced our understanding of the progression of DR. Importantly, OCTA has revealed retinal changes that precede visible DR lesions on traditional examination. OCTA shows promise in predicting the development of visible DR, potentially providing an avenue for preventative measures to manage this sight-threatening condition.

### 4.2. Diabetes Mellitus without Clinical Signs of Diabetic Retinopathy

Researchers have been investigating the potential for OCTA to detect microvascular changes that precede clinical retinopathy. Compared to controls, a notable decrease in VD in the SCP and DCP has been observed in patients with DM without DR [65]. Some studies have emphasised a significant decrease in VD in the DCP more than in the SCP [66]. However, it is worth noting that this observation has not been replicated in all studies [67,68].

Abnormalities in the FAZ have also come under scrutiny in DM patients without clinical DR. For instance, de Carlo et al. used OCTA to demonstrate a significant enlargement in FAZ in these patients compared to people without diabetes [69]. However, the evidence regarding FAZ changes remains inconclusive, with some studies confirming the enlargement [65,70] whilst others reporting no significant correlation [67,68]. The inconsistency of FAZ findings might be due to inherent OCTA artefacts. Therefore, researchers should review the scans carefully and rectify those correctable artefacts, such as SSPiM (Figure 3), decentration (Figure 4), and segmentation artefacts (Figure 6), before entering the data into the analysis.

Interestingly, Dimitrova et al. found a decreased VD of the CC in patients with DM without DR [65]. However, this observation was inconsistent with other studies conducted by Carnevali et al. and Dai et al. [66,67]. Studying the eyes of newly diagnosed diabetic patients, Dai et al. suggested that the CC flow reduction may precede retinal flow changes in the macula, potentially serving as an early marker of microvascular dysfunction [67].

Despite these findings, it remains challenging to establish consistent results across the retinal microvasculature, potentially due to differences in the resolution of OCTA devices, scan protocols, small sample-sized studies, different analytic methods, and interpretation of findings. Ultimately, well-designed studies on OCTA are required to clarify potential preclinical markers to prevent visual loss in DM patients.

### 4.3. Non-Proliferative Diabetic Retinopathy (NPDR)

The regular monitoring of patients with NPDR is essential to prevent progression to PDR, which can potentially lead to irreversible visual loss. Clinical signs such as cotton wool spots, hard exudates, and retinal haemorrhages observed on CFP are less well appreciated on OCTA. However, OCTA offers a unique advantage of visualising IRMA and venous beading that may be missed on CFP.

An area of particular interest is the development of microaneurysms (MA). According to Park et al., MAs have been observed across all three retinal plexuses [71]. Ishibazawa et al. characterised these MA as focally dilated saccular or fusiform capillaries and identified their more frequent origin from the DCP than the SCP using OCTA [72]. This observation is not unique, as other studies have also reported more MA in the DCP compared to the SCP [73,74], suggesting that initial changes in DR may originate from the DCP. However, it is worth noting that OCTA has limitations in detecting all MAs seen on FFA, possibly due to the blood flow rate within these MAs being too slow to detect [73].

Distinguishing severe NPDR from PDR can be difficult due to the subtle appearance of IRMA on CFP, often misinterpreted as NV [75]. As demonstrated by Arya et al., OCTA offers the solution by distinguishing IRMA, which presents as outpouchings from the inner limiting membrane confined entirely to the retina, unlike NV, which exhibits supraretinal flow crossing membrane and posterior hyaloid [76]. This ability to discern IRMA from NV is a valuable contribution of OCTA.

Although not currently included in the grading of DR, non-perfusion areas (NPA) are a notable feature with predictive potential for the progression from NPDR to PDR [77]. As DR progresses, NPA increases. NPA is an indication of vascular damage and retinal ischaemia [72,78]. OCTA is more effective at visualising these areas than FFA, and OCTA may reveal nonperfusion in regions labelled as perfused on FFA [73]. This finding on OCTA may be due to slow blood flow in these areas, making NPA a clinically valuable marker for tracking DR progression. OCTA has detected NPA in all three retinal plexuses [73] and has been linked with IRMA and neovascularisation [79]. Recently, in a two-year longitudinal study involving 122 individuals with NPDR from type 2 DM, Reste-Ferreira et al. used OCTA to identify a significant association between retinal neurodegeneration, thinning of the retinal ganglion cell layer and inner plexiform layer, and an increase in retinal NPA [80]. These findings offer the potential for the development of new early interventions for the management of NPDR. Wide-field OCTAs are required to incorporate these changes into a new classification system for DR.

### 4.4. Proliferative Diabetic Retinopathy (PDR)

FFA is the established method for detecting NV that is not clinically visible [81]. Given that 17% of patients with NPDR advance to PDR [82], OCTA has become a crucial, non-invasive tool for the rapid identification of PDR. This early detection will become essential when new therapies emerge for treating NPA, as initiating timely treatment will likely prevent visual loss from neovascular complications.

NV occurs as a response to retinal ischaemia and plays a significant role in complications such as tractional retinal detachment and vitreous haemorrhage, ultimately leading to visual loss in PDR [83]. Various studies have classified active NV based on its morphology, blood flow on a density map, or vitreous segmentation [83,84,85]. In their cross-sectional study on eyes with untreated PDR, Pan et al. introduced a comprehensive classification system of NV [86]. They utilised OCTA to investigate both blood flow and morphology of NV, analysing 75 NVEs and 35 NVDs across 35 eyes. They suggested a classification of NVE into three distinct subcategories: type 1, the most prevalent, originates from the venous side of NPA; type 2 originates from the capillaries within the NPA; and type 3 stems from IRMA within NPA. Furthermore, they observed that NVD originated from the retinal artery, retinal vein, or choroid.

Biomarkers derived from OCTA have demonstrated the ability to predict the progression of DR [87,88,89]. The biomarkers include eyes with larger FAZ area, lower VD, and lower fractional dimensions (FD) on DCP, which are associated with a higher risk of DR progression. Meng et al. found that the higher the severity of DR classification, the larger the FAZ area and the greater the reduction in visual acuity [90]. Understanding these biomarkers will allow targeted treatment to prevent visual loss.

Not only can OCTA be utilised to predict the progression of PDR, but it can also be used to monitor response to treatment. Motulsky et al. conducted a two-year study utilising OCTA to monitor NVE and NVD progression in eyes with PDR and tracked response to treatment with anti-VEGF and/or panretinal photocoagulation (PRP) [91]. They could objectively monitor the treatment effect by observing a decreased flow within the NV. Similarly, Russell et al. compared OCTA with FFA to evaluate NV before and after PRP treatment [92]. They discovered that OCTA offered a more detailed visualisation of NV progression and regression following treatment. They proposed that OCTA could be the future primary imaging method for NV management. Various studies have used OCTA to monitor PDR treatment outcomes following PRP [59,93,94,95,96,97]. OCTA has unveiled intriguing findings following PRP, including redistribution of the choroidal circulation from the periphery to the macula [95], FAZ area becoming circular [94], and foveal VD increasing [97]. These mounting results demonstrate the potential for OCTA metrics to be used as clinical trial endpoints for interventions in retinal ischaemia and provide personalised treatment plans for patients.

Despite its potential for monitoring the progression of DR, OCTA still faces challenges in providing a uniform and standardised approach to diagnosis and management. As OCTA continues to develop, addressing these limitations, especially artefacts (see Section 3.2), will be crucial to its evolving success as a diagnostic tool for DR.

### 4.5. Diabetic Macular Oedema (DMO)

DMO is a leading cause of vision loss in patients with DM, affecting approximately 18.8 million DM patients globally in 2020 [4]. On OCTA, DMO is characterised by abnormal fluid accumulation within the inner retinal layers at the macula, driven by angiogenesis and inflammation [98]. Importantly, DMO can occur at any stage of DR and progress independently of DR severity [99]. On OCT, DMO is subdivided into four categories: diffuse retinal thickening, cystoid macular oedema (CMO), serous retinal detachment (SRD), and mixed type [100]. All respond to anti-vascular growth factor (anti-VEGF) agents, but non-responders may require other treatment options, such as steroids, due to varying degrees of inflammation [60].

OCTA has been utilised to quantify several biomarkers indicative of DMO. OCTA metrics associated with DMO include lower VD, lower PD, lower FAZ circularity, increased FAZ area, and higher FD in CC [60,101,102]. These are likely to be signs of decreased macular perfusion associated with DMO. Lower VD in the SCP and decreased perfusion in the DCP have also been identified as significant predictors of DMO development [60]. Increased flow deficit at the CC has also been reported to be associated with DMO development [102]. Intraretinal cystoid spaces are seen as round black flow voids on OCTA, with greater visibility in the DCP than SCP [103]. In chronic DMO, these cystic areas are surrounded by NPA [104]. The preference to develop in NPA and reductions in flow and perfusion suggest that ischaemia at a deeper level may precede DMO.

Moreover, MAs within the DCP have also been thought to contribute to the pathogenesis of DMO [105]. Research has suggested that a higher number of MA in the DCP, coupled with a larger FAZ area, leads to poorer response to anti-VEGF treatment [106]. The effectiveness of anti-VEGF treatment in DMO is also determined by the integrity of the DCP and the size of FAZ [107]. Therefore, OCTA-based biomarkers aid in diagnosis and serve as valuable tools for prognosis and monitoring VA in response to treatment, using factors such as FAZ dimensions and VD at multiple capillary plexuses [108,109].

However, OCTA has a set of practical limitations in evaluating DMO, including segmentation errors due to retinal oedema, the impact of cystoid spaces, and inaccuracies related to axial length [110,111]. Efforts are underway to develop algorithms that minimise such artefacts and improve the reliability of OCTA metrics [112]. As technology evolves and more research is conducted, OCTA is expected to have a central role in DMO management.

### 4.6. Diabetic Macular Ischaemia (DMI)

It has not yet been determined whether DMO triggers DMI or vice versa; however, the introduction of OCTA has led to increased research interest in DMI, raising questions about whether there should be revisions to the classification of DR [7].

DMI is characterised by the enlargement of the FAZ and retinal capillary loss in the parafovea [113]. The prevalence of DMI varies in the literature, with reports as high as 77% in patients with established PDR [114]. The true prevalence of DMI may be underreported as key DMI studies used FFA, where the scattering of fluorescein might prevent researchers from precisely deciphering the FAZ [115,116]. It is known now that the grading of DMI using OCTA is comparable to standard FFA [117]. DMI could ultimately cause irreversible visual loss with no known prevention or treatment strategies. It has been recognised as a risk factor for the progression of DR [114] and can be used as a prognostic factor for the treatment of DMO [106].

Neuronal changes in DMI are characterised by DRIL and ellipsoid zone (EZ) loss [118]. The presence of DRIL is associated with multilevel retinal capillary NPA [119] and is considered a significant predictor for future VA outcomes in eyes with DMO, especially when centrally located [120]. Recent research by Tsai et al. focused on the association between DMI, DRIL, and EZ [118]. Their findings revealed that DRIL is more prevalent than EZ loss in patients with DMI. Notably, eyes with DRIL alone showed worse functional and microvascular metrics than eyes with EZ loss alone, while eyes with both DRIL and EZ loss exhibited the poorest functional outcome. As DMI severity increases with the presence of DMO, DRIL, or both, the impact on visual function becomes more pronounced [7]. FAZ area, EZ disruption, and DRIL length have all been found to correlate with poor visual acuity [121]. Despite these observations, the pathogenesis and effect of DMI have not been fully understood, and further research with OCTA is needed to correlate these changes in visual function.

Independent analysis of the retinal capillary beds using OCTA is crucial in understanding DMI. The first cross-sectional analysis of the use of OCTA assessing NPA at the level of SCP and DCP found that, out of the 1107 DM eyes, DCP-DMI was more prevalent than SCP-DMI [122]. DCP-DMI has been found to have a stronger correlation to functional deficit than SCP-DMI [109]. Interestingly, DMI can still be observed in eyes without DR at the level of SCP and DCP, suggesting that ischaemia may occur before any clinical signs of DR are present [122]. The suggestion means that the use of OCTA may be clinically important to first identify DMI prior to them exhibiting any signs of DR. DMI has also been associated with decreased capillary perfusion and VD in the SCP, DCP, and CC [117,123,124]. Bradley et al. devised an OCTA-graded scale for DMI and found a high degree of intergrader agreement for DMI grades when evaluating OCTA images [117]. These ischaemia grades from the SCP and CC had a statistically significant relationship with decreased flow indices (version 21, SPSS).

Cheung et al. have classified DMI into three distinct clinical phenotypes using OCTA: generalised DMI, predominant-DCP ischaemia, and predominant-SCP ischaemia [7]. These were proposed to understand the variability in the degree of visual impairment observed among individuals with DMI. They suggested potential options for preventing and treating DMI based on the phenotype and recommended further research with the help of OCTA to facilitate future therapeutic management.

### 4.7. Pre-Diabetes

Research into pre-diabetes is becoming increasingly popular, previously overlooked due to its absence as a recognised category in the classification of DR. Pre-diabetes is a condition in which blood sugar levels are higher than normal but do not reach the threshold for type 2 diabetes diagnosis [125]. In 2021, the global prevalence was estimated at 298 million, a number which is projected to rise to 587 million in 2045, according to the International Diabetes Federation [126,127]. A recent meta-analysis found that the prevalence of retinopathy in pre-diabetes is 7.1%, and pre-diabetes was found to be associated with early retinopathy changes [128].

Several studies have utilised OCTA metrics to assess the microvascular changes induced in patients with pre-diabetes. Arias et al. studied 23 eyes of patients with pre-diabetes and compared them with 20 eyes of healthy controls [129]. They found that the PD and VD in both the SCP and DCP were reduced in patients with pre-diabetes compared to healthy controls. The finding is similar to Kirthi et al., who also found that patients with pre-diabetes had a decreased parafoveal VD in the SCP and DCP compared to normoglycemic controls [130]. A cross-sectional study found that only the VD in the SCP was decreased in patients with pre-diabetes compared to controls, whereas VD in the DCP was decreased in the diabetic group [131]. However, Xu et al. found that the VD and vessel area density were similar between patients with pre-diabetes and controls [132].

Although Xu et al. reported FAZ area to be significantly enlarged in patients with pre-diabetes [132], most studies have found no significant change in FAZ area and circularity in pre-diabetic patients compared to controls [129,130,133,134]. Ratra et al. did, however, find that logmarBCVA was positively correlated with FAZ area in the pre-diabetic group [134]. Zaagst et al. evaluated patients with type 2 diabetes, pre-diabetes, and controls using multifocal electroretinogram (mfERG) and OCTA; they concluded that the FAZ area was significantly negatively correlated with mfERG amplitudes averaged over both the posterior pole and fovea, thereby postulating that increased glucose level is associated with an enlarged FAZ and also decreased mfERG amplitude [135]. FAZ area was also found to be positively correlated with age in all groups.

One study looked at a new parameter, microangiopathy, on OCTA. El Sawy et al. were able to detect retinal microangiopathy in 36.4% of pre-diabetics, capillary NPA in 53.8%, and the disorganisation of the SCP in 56.8% of pre-diabetic patients, compared to only 10.6% detected in CFP [136].

It has been shown that early retinal microvascular changes can develop in pre-diabetes before the onset of overt diabetes. Although there is no consensus among studies, OCTA could be useful in screening patients with pre-diabetes, detecting changes in the FAZ or VD, and perfusion in the SCP and DCP before any clinical signs manifest. Low reproducibility and variations across different OCTA devices limit current clinical use. Future prospective studies are needed to establish the correlation between the OCTA metrics and functional outcomes such as BCVA and further investigate early microvascular impairment in patients with pre-diabetes.

## 5. Different Scanning Protocols in Optical Coherence Tomography Angiography

Different OCTA scan protocols are used for various clinical and research purposes. As all scan protocols utilise 304 × 304 B scans, it results in a higher scan density for the 3 × 3 mm scan density than the 6 × 6 mm scans [137]. Consequently, the 3 × 3 mm protocol offers a more distinct delineation of findings because of the higher resolution.

Ho et al. compared 3 × 3 mm and 6 × 6 mm scans to evaluate NPDR [137]. They reported that a 3 × 3 mm scan provided superior delineation of the FAZ and demonstrated improved remodelling compared to the 6 × 6 mm scan, primarily attributed to its higher scan density. On the other hand, the 6 × 6 mm scan exhibited greater sensitivity in detecting microaneurysms when compared to the 3 × 3 mm scans, primarily due to their large scan area.

Thompson et al. demonstrated a statistically significant increase in macular VD, measured within a 6 × 6 mm field-of-view using OCTA in a cohort of 20 diabetic patients without clinical signs of DR [61]. It is suggested that the increase in VD can potentially signify a progression of subclinical disease among diabetic patients, possibly marking the initial stages of underlying choroidal tissue hypoxia.

Hirano et al. compared three different scan sizes (3 × 3 mm, 6 × 6 mm, and 12 × 12 mm) to assess the differences in retinal parameters between patients with DR and healthy individuals [138]. The study found that regardless of the retinal slab and scan size used, patients with DR had significantly lower PD, VLD, and FD than the control group. Furthermore, lower PD, VLD, and FD were consistently identified in patients with DR across all retinal layers and scan sizes. Significant differences in all metrics were found between NPDR eyes with and without DMO in the deep retinal layers of the 3 × 3 mm SS-OCTA images. However, this difference was not observed to the same extent in PDR eyes for the superficial retinal layers or full-thickness retina. Additionally, these differences were not seen in the 6 × 6 mm and 12 × 12 mm SS-OCTA images.

Santos et al. obtained SS-OCTA images from both the 3 × 3 mm and 15 × 9 mm protocols. They found that retinal capillary closure could identify NPDR severity progression [139]. This phenomenon was more readily visible in the perifoveal area during the initial stages, whereas the involvement of the retinal mid-periphery became prominent only in the advanced stages of retinopathy.

In addition to other metrics, Mendes et al. included an assessment of abnormal intercapillary spaces (AIS) in their evaluation of vessel closure (VC) across three ETDRS severity groups [55]. They reported that AIS could effectively distinguish between eyes with DR and the control group; however, this metric was associated with a relatively high variation subject to the binarised *en-face* slab image quality.

Most recently, besides similar findings with VD, Yang et al. evaluated blood flow density with 3 × 3 mm windows [122]. It is reported that central retinal blood flow density (CRBFD), inner retinal blood flow density (IRBFD), wide-field RBFD, and outer RBFD reduced significantly in patients with DR.

While prior investigations indicated a reduction in the FAZ area, it is noteworthy that a recent study suggested that the FAZ may not exhibit consistent alterations in the earliest stages of DR [139]. Santos et al., using 3 × 3 mm scans, reported that eyes with preclinical retinopathy demonstrated early and predominant occurrences of retinal capillary closure, leading to an apparent decrease in VD and PD. These initial microvascular changes were primarily localised within the SCP. These findings indicate that the initial microvascular changes within the retina in T2DM are characterised by capillary closure or a reduction in blood flow, specifically within the SCP. This proposal also corroborates the results reported by de Carlo et al. [69].

## 6. Recommended Future Research Directions

OCTA has demonstrated its effectiveness in visualising the retinal microvasculature and aiding in the early detection and management of all stages of DR, including DMI and pre-diabetes. Despite these advantages, a major limitation has been the absence of a standardised methodology for comparing quantitative indices in OCTA. This lack of standardisation hinders the development of a unified database. Recently, there have been efforts to establish a standardised nomenclature to describe various OCTA parameters in assessing DR severity [140,141].

To address this challenge, researchers need to establish correlations between various OCTA metrics, such as FAZ area and VD, and functional measures of visual function, such as visual acuity. Creating a formula to convert different metrics from various OCTA machines will enable better use of OCTA in a clinical setting and facilitate greater interchangeability between various OCTA devices [142]. Factors such as age [143,144], sex [145,146], HbA1c levels [147], and ethnicity [148] influencing OCTA measurements emphasise the need for matched controls in the database. Novel algorithms have been developed, such as variable interscan time analysis (VISTA), which uses blood flow speed to map retinal vasculature [149]. One can hope these algorithms will be interchangeable with different OCTA devices.

Enhancing the quality of OCTA would be beneficial for more effective DR detection. Recently, there has been a focus on developing new techniques to improve the resolution and flow sensitivity of OCTA images without prolonging the acquisition time to ensure patient comfort. Currently, OCTA devices have a restricted field of view with low resolution, limiting their usefulness as a diagnostic tool [20,150]. Newer OCTA devices that offer better resolution with a wider field of view are being evaluated to overcome these limitations.

Artificial intelligence (AI) and deep learning algorithms provide cost-effective and more efficacious methods to improve clinical outcomes in DR [151]. For instance, Sandhu et al. designed a computer-aided diagnostic system which used three OCTA features, including VD, blood vessel calibre, and FAZ size, to diagnose and classify patients with NPDR, demonstrating an overall accuracy of 94.3% [152]. Similarly, Alam et al. tested a super vector machine classification model for computer-aided NPDR image classification using six qualitative OCTA features. This model achieved 94.4% and 42.96% accuracy for control versus disease and control versus mild NPDR, respectively [153]. Researchers also used a machine learning-based multitask OCTA classification model to differentiate normal conditions from ocular and distinguish between various ocular diseases [154]. Using this model, they achieved 92% specificity in distinguishing sickle cell retinopathy from DR. Such AI-based systems can be used as a more affordable evaluation tool for disease screening, especially in rural settings that may lack expert eye care providers. Larger population trials are essential to evaluate the potential of AI fully [142].

Another area of interest is the development of portable OCTA systems for enhanced flexibility in imaging patients in supine positions, paediatric age groups, and those residing in remote areas without the infrastructure for setting up a conventional OCTA device [155]. Rank et al. developed the first chip-based OCTA system, which is a big step towards developing smaller and more compact devices that may further promote the usefulness of OCTA as a screening tool across diverse sectors [156].

Finally, while current research primarily focuses on evaluating posterior segment abnormalities, OCTA’s potential in assessing the anterior segment is gaining interest. It involves using an adapter lens with manual equipment adjustment for imaging the anterior segment. This development allows early detection of iris neovascularisation and other vascular abnormalities that can occur as a sequela in PDR [157,158]. However, this technique is still in its early stages and requires further evaluation as most currently used software is designed for the posterior segment and can result in inaccuracies when used for studying the anterior segment [159].

OCTA holds significant potential for enhancing clinical outcomes in DR, provided ongoing research addresses its limitations and explores new applications.

## 7. Conclusions

The global increase in DM emphasises the urgency for effective diagnosis and management of DR. Over the last few years, OCTA has proven to be a fast and non-invasive alternative to FFA in the diagnosis and management of DR. It provides three-dimensional, depth-resolved images of the retinal vasculature, adding depth to our understanding of DR.

OCTA also provides a detailed analysis of the different stages of diabetic retinopathy. With its ability to detect subtle microvascular changes, OCTA shows promise in identifying retinopathy in its earliest stages, including pre-diabetes and diabetes without clinical retinopathy. An updated classification for different stages of DR would be beneficial, given the increasing evidence around DMI and pre-diabetes.

Efforts are being made to establish a unified database, addressing the lack of standardisation in the field. AI could be used as a cost-effective solution for screening. Standardisation and further research are needed before OCTA is used routinely in clinical practice. The future of OCTA as a diagnostic tool for DR is promising, offering valuable insights into the disease to help prevent visual loss.

## Figures and Tables

**Figure 1 diagnostics-14-00326-f001:**
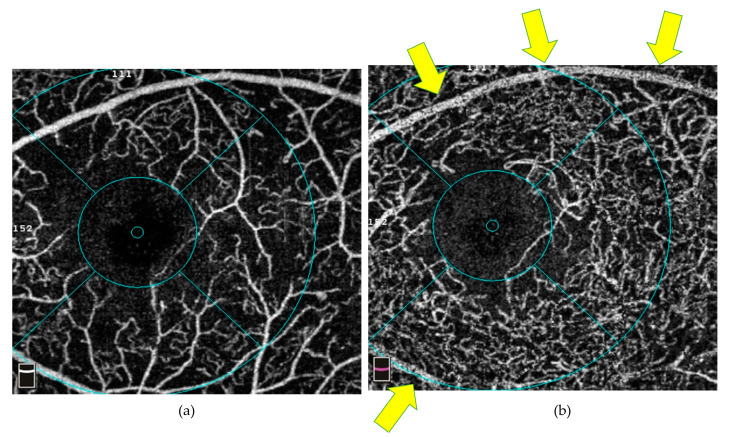
An example of an OCTA projection artefact from a 3 × 3 mm macular scan. (**a**) Major blood vessels seen in the SCP scan and (**b**) major blood vessels (arrows) seen as a projection artefact in the DCP scan. Abbreviations: DCP = deep capillary plexus; OCTA = optical coherence tomography angiography; SCP = superficial capillary plexus.

**Figure 2 diagnostics-14-00326-f002:**
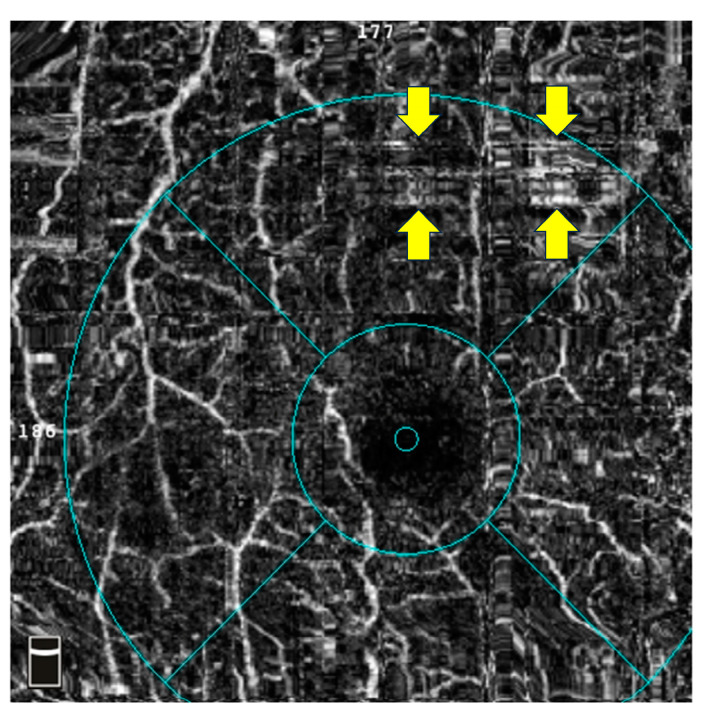
An example of OCTA motion artefact from a 3 × 3 mm macular scan. Motion artefacts can create false positive signals (arrows indicate the most obvious ones), which might result in erroneous vessel density calculation and wrongly delineated foveal avascular zone contour. Abbreviations: OCTA = optical coherence tomography angiography.

**Figure 3 diagnostics-14-00326-f003:**
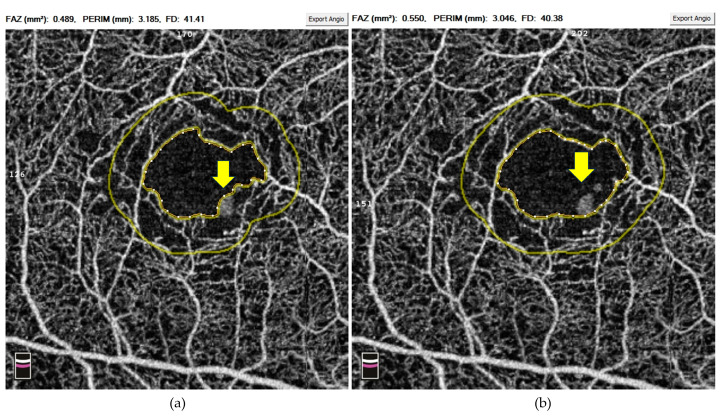
An example of SSPiM-related false positive signal and associated issues within the 3 × 3 mm macular region on OCTA. (**a**) FAZ area and perimeter containing scattering particles in motion (arrow); (**b**) after manual correction, the scattering particles in motion (arrow) were included, therefore, there was an increase in the FAZ area and perimeter. Abbreviations: FAZ = foveal avascular zone; OCTA = optical coherence tomography angiography; SSPiM = suspended scattering particles in motion.

**Figure 4 diagnostics-14-00326-f004:**
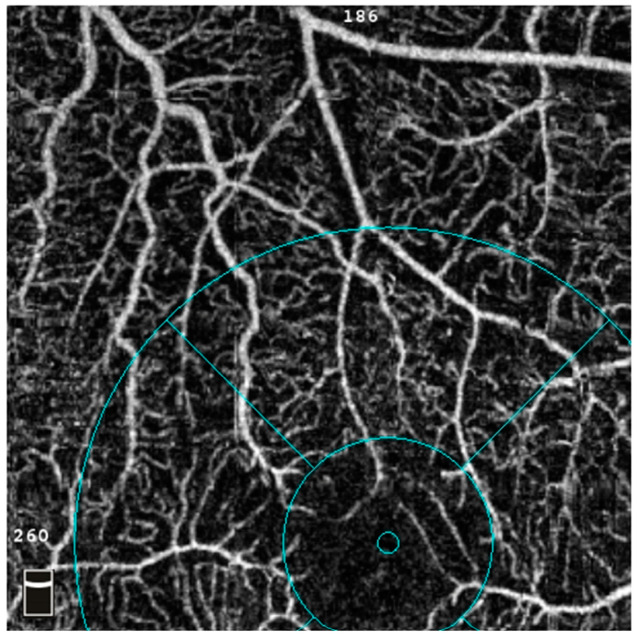
An example of OCTA decentration artefact on a 3 × 3 mm macular scan. The FAZ is downward displaced, resulting in a wrong interpretation of the size of the FAZ. In addition, the whole image and parafoveal VD readings are incorrect, which should be carefully excluded before pooling for final analysis. Abbreviations: FAZ = foveal avascular zone; OCTA = optical coherence tomography angiography.

**Figure 5 diagnostics-14-00326-f005:**
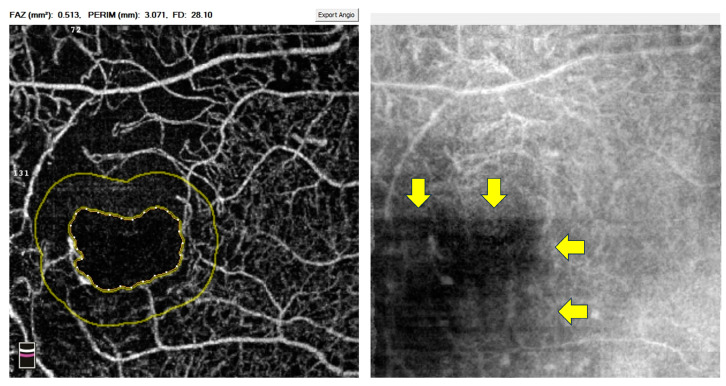
An example of OCTA shadow artefact on a 3 × 3 mm macular scan. There is an area of false negative signals (indicated by arrows) on the *en-face* OCTA image on the right-hand side. This shadow artefact can severely reduce the capillary plexus signals in the same area on the reconstructed AngioVue image on the left. Abbreviations: OCTA = optical coherence tomography angiography.

**Figure 6 diagnostics-14-00326-f006:**
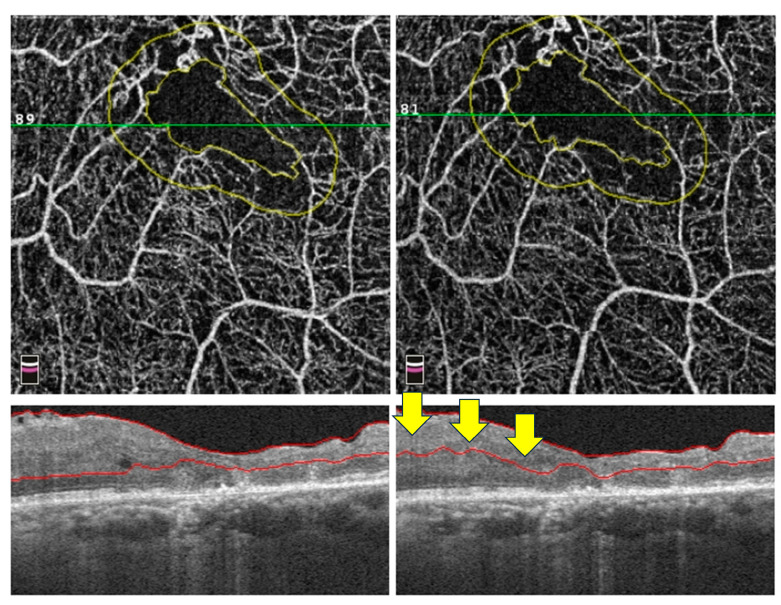
An example of OCTA segmentation artefact on a 3 × 3 mm macular scan. The B-scan on the right-hand side shows erroneous segmentation of the outer plexiform layer’s outer border (as indicated by arrows). In contrast, the B-scan on the left-hand side from the same eye is more accurately segmented. Abbreviations: OCTA = optical coherence tomography angiography.

**Figure 7 diagnostics-14-00326-f007:**
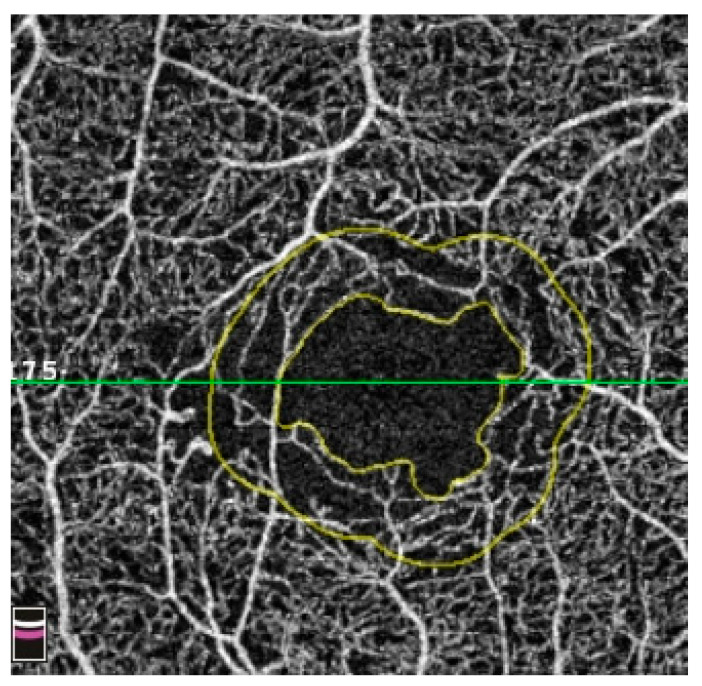
A 3 × 3 mm macular OCTA image showing signs of DMI, characterised by an enlarged and irregular FAZ with loss of perifoveal capillaries. This picture was taken six months later from the same eye in Figure 3, demonstrating an enlargement of the FAZ area. Abbreviations: DMI = diabetic macular ischaemia; FAZ = foveal avascular zone; OCTA = optical coherence tomography angiography.

**Table 1 diagnostics-14-00326-t001:** Comparison of OCTA versus FFA, adapted from Tan et al. [23].

OCTA	FFA
Non-invasive	Invasive—dye required, risk of anaphylaxis
Rapid	Time-consuming
Distinguishes retinal layers	Unable to distinguish retinal layers
3D image	2D image
Superior high image resolution	Lower image resolution
Quantitative data	Qualitative data
Artefacts may cause interpretation errors	Artefacts less common
Smaller field of view	Wider field of view
Able to detect flow but not leakage	Able to detect flow and leakage
No validated technique	Validated technique
Easier to conduct on patients with poor venous access	Difficult to conduct on patients with poor venous access
More suitable for imaging in children	Less suitable for imaging in children

Abbreviations: FFA = fundus fluorescein angiography; OCTA = optical coherence tomography angiography.

**Table 2 diagnostics-14-00326-t002:** Different optical coherence tomography angiography (OCTA) quantitative metrics [7].

OCTA Metric	Description
Foveal Avascular Zone (FAZ) area	Measurement of the FAZ size in mm^2^
Foveal Avascular Zone (FAZ) circularity	Describes how circular FAZ appears at the fovea
Fractal Dimensions (FD)	Irregularity of blood vessel pattern
Intercapillary Spaces (IS)	Space between adjacent capillaries
Non Perfusion Area (NPA)	Area of absent blood flow
Perfusion Density (PD)	A similar concept of VD used by Zeiss
Vessel Density (VD)	Proportion of vessel area with blood flow
Vessel Skeleton Density (VSD)	Density of binarised vessel network

## Data Availability

No new data were created.

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
