# Peer review of "Optical Coherence Tomography Angiography as a Diagnostic Tool for Diabetic Retinopathy"

_diagnostics, 2024, doi:10.3390/diagnostics14030326_

Round 1

Reviewer 1 Report (Previous Reviewer 2)

Comments and Suggestions for Authors

Thanks the authors for their responses to my previous comments. Additional minor comments are:

Please add relevant references for line 234-243.

Figure captions: are the units of the size mm by mm?

Comments on the Quality of English Language

N/A.

Author Response

Thank you very much for taking the time to review this manuscript. Please find the detailed responses attached and the corresponding revisions in track changes in the re-submitted files.

Reviewer 2 Report (New Reviewer)

Comments and Suggestions for Authors

The authors present a review paper on the use of optical coherence tomography angiography (OCTA) in the early diagnosis and management of diabetic retinopathy (DR). This is a very comprehensive review on a very relevant topic with high social impact. The paper is well-written and well organized. It clearly deserves to be published.

I would like to praise the section on artifacts, which I think is very clear and very relevant. The artifacts in angiograms were one of the factors that most delayed the acceptance of OCTA and despite the significant effort that has been made by manufacturers to reduce them, it is still essential that clinicians are aware of their existence and their cause, and how they can affect angiograms. I think this section is very well done.

I have just a few minor suggestions that, in my opinion, may improve the paper.

The authors chose not to discuss the fundamentals of OCTA and did not include a detailed description of the aspects related to the instrumentation and the algorithms used to monitor the movement of red blood cells over time. This is a perfectly justifiable choice given the scope of the review, which is focused on topics that have an impact on the use of OCTA in a clinical environment, in the context of DR, and on the correct analysis of output angiograms. However, I think it would be justified to include a paragraph stating this option and pointing to references in the literature where the reader could find a more detailed description and discussion of the instrumentation of OCTA systems and the most used algorithms used, namely the split spectrum amplitude decorrelation technique.

One advantage over FFA that was not explicitly mentioned, is the higher suitability and ease of OCTA for imaging children and people with poor venous access.

The main limitation of OCTA lies on not being able to detect leakage. In my opinion, as long as this limitation exists, OCTA will ever fully replace FFA. There are ongoing research efforts to overcome this limitation and a few papers were already published addressing potential approaches. I believe that the review paper would improve by including a brief reference to these efforts.

Author Response

Thank you very much for taking the time to review our manuscript. Please find the detailed responses attached and the corresponding revisions in track changes in the re-submitted files.

This manuscript is a resubmission of an earlier submission. The following is a list of the peer review reports and author responses from that submission.

Round 1

Reviewer 1 Report

Comments and Suggestions for Authors

Interesting topic but it has been widely investigated with literature and systematic review.

Authors are not bringing anything news and they don’t fill any gap of knowledge. In this actual form it is very similar to these publications

DOI: 10.1038/s41433-020-01233-y

DOI: 10.1111/aos.13859

DOI: 10.1111/aos.13859

DOI: 10.1007/s10792-018-1034-8

DOI: 10.3390/biomedicines10010088

DOI: 10.1016/j.jcjo.2019.02.010

Authors should work on their originality and strengths

- 2. Principles of OCT-A: it would be interested to bring some information regarding choriocapillaris plexus. It is innovative and it is available with recent OCT-A device

- interesting paragraph on artefact, i would recommend to present more samples of OCT-A in order to illustrate different artefact motion, projection processing, FP and FN blood flow. with FAZ segmentation and OCT-A metrics. It would also illustrate how metrics could be influenced by artefacts.

Reviewer 2 Report

Comments and Suggestions for Authors

In this manuscript, Wijesingha et al. reviewed applications of OCTA for Diabetic Retinopathy (DR) diagnosis. OCTA principles, comparisons with current standard FFA, artifacts, and applications in clinics are presented. Please see my comments listed below.

·         Line 84: Please correct spelling of ‘threating’.

·         Table 1: How is leakage detection related to DR diagnosis and why does that make OCTA non -superior compared to FFA?

·         Section 2.2.1: As stated in the OCTA principles section, OCTA monitors motion of red blood cells within vessels. So, one main reason for OCTA’s false negative performance is this intrinsic principle. For instance, the cell free layer of large vessels won’t show up in OCTA, and this is why OCTA vessel diameter is smaller than its actual value. In addition, OCTA is not able to detect plasma skimming vessels (no blood cell traveling in the vessel). Please add this point in the artefacts section.

·         Figure 1: what is the image size? Please add a scalebar or state in the caption to show this.

·         Figure 2: I am interested to know, what would be the scattering particle in the image? Also, what does the larger yellow line represent? What were the correction criteria of the larger yellow contour?

·         The manuscript has included comprehensive literature studies, and for a specific disease or symptom, different observations might be presented. For instance, line 433-442, whether FAZ area is enlarged in patients with prediabetes or not, remains unclear. Can the authors comment on this? What would be potential reasons for these different observations?

Comments on the Quality of English Language

Please check spelling and usage thoroughly through the manuscript.